# Mechanism of Fibrosis Induced by *Echinococcus* spp.

**DOI:** 10.3390/diseases7030051

**Published:** 2019-08-12

**Authors:** Fuqiu Niu, Shigui Chong, Mingqun Qin, Shenmei Li, Riming Wei, Yumin Zhao

**Affiliations:** 1Department of Parasitology, School of Basic Medicine, Guilin Medical University, Guangxi 541004, China; 2School of Stomatology, Guilin Medical University, Guangxi 541004, China; 3School of Biotechnology, Guilin Medical University, Guangxi 541004, China; 4Department of Nursing teaching and Research, Guilin Medical University, Guangxi 541004, China

**Keywords:** *Echinococcus*, hepatic fibrosis, miRNA, TGF-β

## Abstract

Infection with *Echinococcus* spp. causes fibrosis in various vital organs, including the liver and lungs. Hepatic fibrosis is a pathological feature of *Echinococcus* infection that destroys normal liver tissue, leading to jaundice, cholecystitis, portal hypertension, etc. Severe *Echinococcus multilocularis* infections lead to liver failure and hepatic encephalopathy. The formation of peripheral fiberboards around the metacestode is a major reason as to why antiparasitic drugs fail to be effectively transported to the lesion site. Studies on the mechanism of hepatic fibrosis caused by *Echinococcus* are important for treatment in patients. Recent studies have focused on miRNA and TGF-β. More recent findings have focused on the generation of collagen fibers around the metacestode. In this review paper we focus on the mechanism by which the *Echinococcus* parasite induces fibrosis in liver and some other organs in intermediate hosts—animals as well as human beings.

## 1. Introduction

*Echinococcus* spp. includes *E. granulosus*, *E. multilocularis*, *E. vogeli*, *E. oligarthrus* and *E. shiquicus* [1]. Patients are only rarely infected with *E. vogeli*, *E. oligarthrus* or *E. shiquicus* [2,3,4]. In this paper, we focus on *E. granulosus* and *E. multilocularis*. The eggs released from adult *E. granulosus* pollute pastures. If the water source is consumed by the intermediate host (cattle, sheep and other herbivores or humans), the oncosphere enters the duodenum through the upper digestive tract. Under suitable conditions, the oncosphere breaks out and enters the intestinal wall, which then reaches the liver and the lung mainly through blood flow, subsequently developing into metacestodes within 3–5 months. Metacestodes of *E. granulosus* grow slowly and last for several years or decades, mainly in livers or lungs. When dogs, as the definitive host, accidentally eat organs containing metacestodes, protoscoleces are released into the small intestine, and develop into adults to complete their life cycle [5]. The life-cycle of *E. multilocularis* is similar to that of *E. granulosus*. Adults live in the small intestine of *Vulpes*, *Canis* and *Felis*, while larvae (metacestodes) migrate in the liver of rodents. As an unsuitable intermediate host, humans become infected by accidental consumption of contaminated food with eggs containing oncospheres [6]. The life-cycles of *E. granulosus* and *E. multilocularis* are shown in Figure 1 and Figure 2, respectively [6,7]. *E. granulosus* and *E. multilocularis* cause cystic and alveolar echinococcosis, respectively [8]. Both mainly occur in the liver and promote hepatic fibrosis [9,10]. One possible molecular mechanism of *E. multilocularis* organ tropism towards the liver is that the *E. multilocularis* tends to migrate to organs with high concentrations of insulin. Indeed, the highest concentrations of insulin in mammals occur at the junction between the portal vein and the liver parenchyma. As an evolutionarily conserved signaling pathway, insulin from the host could interact with receptors of *E. multilocularis* as ligands. The effect of insulin signaling pathway in *E. multilocularis* including generation of metacestodes, differentiation of *E. multilocularis* oncospheres towards the metacestode stage, and ‘re-differentiation’ of *E. multilocularis* protoscoleces towards the metacestode stage [7].

*E. granulosus* lesions differ from *E. multilocularis* as a pathological manifestation. In sheep infected with *E. granulosus*, the surface of the liver becomes uneven, and the number of protruding vesicles increases. The liver parenchyma has several metacestodes (Figure 3B). A large amount of fluid accumulates in the daughter cysts of metacestodes. In addition to infertile cysts, hydatid sands are visible (Figure 3C). Many protoscoleces can be imaged on slide smears, and some metacestodes become calcified. Our pathological sections and staining of the metacestodes showed that the laminated layer consists of a uniformly red-stained colloidal substance, and the germinal layer is inside the laminated layer, which is composed of irregularly arranged single cells (Figure 4). In another experiment, rodents were infected with *E. multilocularis*, and the sections prepared from masses in the liver. Pale yellow or white vesicle-like cells could be observed in these sections by gross examination. Multiple metacestodes were connected and aggregated, round or oval, and contained transparent vesicle fluid or gelatinous substances (Figure 3A). A large number of protoscoleces could be imaged under the light microscope. Histological sections and staining showed that numerous metacestodes were clustered into groups. A large number of laminated layers and germinal layers could be detected in these clusters and a majority of germinal layers could be found fallen off these laminated layers. The laminated layer was bent and acellular, and curled up in the vesicle cavity (Figure 5) [11].

Cystic echinococcosis is globally distributed, and high prevalence areas were concentrated in pastoral areas of Eurasia, Africa, America and Oceania. In China, Cystic echinococcosis is prevalent in Qinghai-Tibet Plateau, Inner Mongolia Plateau, Yunnan-Guizhou Plateau and Loess Plateau. Alveolar echinococcosis is distributed in the rural areas located in the northern hemisphere, showing a local epidemic trend. In China, alveolar echinococcosis is distributed in Western Sichuan, Qinghai, Xinjiang, Tibet, Gansu and Ningxia [12]. Alveolar echinococcosis and cystic echinococcosis cause harm to human health and animal husbandry [13,14].

Hepatic fibrosis leads to an imbalance of deposition and degradation of the extracellular matrix (ECM) [15]. The over-deposition of ECM leads to cirrhosis and the loss of liver function. Collagen is the main component of the ECM. Type I and type III collagen account for 60% of liver collagen and are closely related to liver fibrosis. Type I and type III collagen increase during the early stages of hepatic fibrosis, and further increase are seen with the aggravation of hepatic fibrosis [16].

Studies on the pathogenesis of liver fibrosis include histopathological changes, toxicology, cytokines and molecular structures. The activation of hepatic stellate cells (HSCs) is central link to the development of hepatic fibrosis. Various factors (cytokines, growth factors, and enzymes) regulate HSCs and fibroblasts in the liver, leading to their biological dysfunction and subsequent liver fibrosis.

## 2. Methodology

We collected information from PubMed and CNKI. The key words used in the PubMed database were ‘fibrosis’ and ‘*Echinococcus*’. We used the Chinese characters ‘fibrosis’ and ‘*Echinococcus*’ as the key words in the CNKI database. The time range we applied in both two databases was ‘2010–2019′. The references published before 2010 were found in references published during 2010–2019. Some papers were also not used because their topics were mainly about other kinds of pathological changes, or the fibrosis was induced by other parasites. Figure 1 and Figure 2 were taken from the PubMed database by searching for the key words ‘*Echinococcus*’. Figure 3, Figure 4 and Figure 5 were obtained from our investigations. From August 2004 to September 2007, in the Gansu province, Tibetan Autonomous Prefecture, we performed 2047 necropsies on animal hosts of *E. granulosus* and *E. multilocularis*, including 1021 sheep, 331 rodents, 634 yaks and 61 domesticated sheepdogs for pathological and epidemiological studies. We captured rodents by rat clips and sticky rat boards, and sources and species were recorded. After dissection, metacestodes were identified by gross examination and saved. The size, number, location and nature of metacestodes were recorded. Pathologic examinations, isolation and identification of metacestodes were performed in our laboratory. Pathological sections of metacestodes were prepared according to routine procedures. Figure 3A and Figure 5 are from rodents infected with *E. multilocularis*. We collected random samples of sheep from slaughterhouse and dissected and performed routine examinations by routine laboratory techniques. The samples in Figure 3B,C and Figure 4 were from sheep infected by *E. granulosus*. Figure 1 and Figure 2 were the results of our epidemiological surveys.

## 3. Mechanisms of Hepatic Fibrosis Induced by *Echinococcus*

### 3.1. MiRNAs Play an Important Role in the Hepatic Fibrosis Induced by Echinococcus

Recent studies have shown that miR-29 regulates hepatic fibrosis and is the regulatory hub for TGF-β and NFκB signaling in HSCs [17,18,19]. Overexpression of microRNA-19b inhibits TGFβRII, inhibiting TGF-β signaling and reducing COL1A1 and α-SMA expression [20]. Select miRNAs promote the migration and activation of HSCs, including miRNA-16, miRNA-195, miRNA-335 and miRNA-181b [21,22,23,24]. Other miRNAs, including miRNA-19b, miRNA-150 and miRNA-29, inhibit HSC activation and ECM synthesis by targeting fibrogenic factors [20,25,26].

Bi and colleagues found that the excretion/secretion products from metacestodes of *E. multilocularis* upregulate miR-133a and COL1A1, and activate TGF-β/smad signaling to activate HSCs in a mice model [27].

It has been shown in the hepatic tissues of alveolar echinococcosis patients that the expression of α-SMA, COL1A1, COL3A1 and TGFβ II in the adjacent region of the lesions was higher than that of the distal regions [26]. While the expression of miRNA-19b in the lesions was lower in the distal tissues, a negative correlation to COL1A1 was noted [28].

Zhang and colleagues suggested that *E. granulosus* may inhibit miR-19 liver expression and promote fibrosis through increases in TβRII, the activation of HSCs and ECM production. The experiments were performed with *E. granulosus*-infected human livers [29].

### 3.2. TGF-β Plays an Important Role in the Hepatic Fibrosis Induced by Echinococcus

TGF-β is the most important signal transduction pathway during hepatic fibrosis. As the most potent fibrogenic factor, TGF-β is important during HSC transdifferentiation [30,31]. Inhibiting the expression of the TGF-β receptor, TGF-β RII, in particular, inhibits the activation of HSCs [32,33,34,35].

In addition, exogenous hydatid fluid of *E. granulosus* from alveolar echinococcosis patients were added to cultured LX-2 cells (a type of HSCs) showed that the hydatid fluid promotes cell proliferation through promoting G0/G1 to S phase transition. Exogenous hydatid fluid of *E. granulosus* could promote the expression of α-SMA, COL1A1, COL3A1 and TGF β RII at the mRNA and protein levels [28].

The downstream Smads of TGF-β1 and its receptors were markedly expressed in periparasitic infiltrates and hepatocytes, which were distant from alveolar echinococcosis lesions [26]. Fibrosis was significant at 180 days post-infection (P.I.) in the periparasitic infiltrates and was present in the liver parenchyma, distant from the lesions. Over the time course of infection, TGF-β1 expression correlated with the CD4/CD8 T-cell ratio previously described as a hallmark of alveolar echinococcosis severity. This suggests that TGF-β plays an important role in alveolar echinococcosis both in immune tolerance against the parasite and during liver fibrosis. These experiments were performed in both a *murine* alveolar echinococcosis model and in alveolar echinococcosis patients [36].

Other investigators have reported that there are no significant changes during the early stages of *E. granulosus* infection in the levels of TGF-β1, but significant increases in the levels of TGF-β1 during the middle and late stages of infection were observed. RT-PCR showed that, when compared with the control group, TGF-β1 mRNA levels were low during the early stages of infection, and significantly increased at day 30 P.I. These remained at high levels until day 270. These experiments were performed in mice model [37].

### 3.3. Collagen Fiber around the Metacestodes of E. granulosus

Histopathological observations from hematoxylin-eosin staining showed that the fibrocyst wall around the metacestode cyst was divided into two layers with clear boundaries. The fibrous layer near the metacestode (inner layer) was a dense, disordered fibrous tissue, often hyaline, lacking blood vessels. Fragments from the laminated layer of the metacestode, epithelioid cells and multinucleated giant cells infiltrated the inner layer, which was surrounded by monocytes, lymphocytes and neutrophils. The fibrous layer near the side of the liver tissue (outer layer) consisted of proliferative fibrous membranes between the liver tissue and the inner layer, which was loosely or compactly arranged. Small blood vessels, bile duct nerves, lymphatic vessels and lymphocytes were scattered in the outer layer. Immunohistochemical staining showed that the expression of type I, III and IV collagen was localized in the outer layer of the fibrocyst wall around the metacestode, showing a brown-yellow sheet distribution, which was occasionally expressed in the inner layer, mostly in a light yellow scattered distribution. The results were obtained from 40 cystic echinococcosis patients [38]. The authors also reported that the mRNA levels of TGF-β 1 and TNF-α in the outer layer were higher than the inner layers [39].

It is proposed that the inner layer is generated through *Echinococcus*–host immune reactions and foreign body granuloma formation. The outer layer is generated through hepatic fibrosis, induced by the compression of expending metacestodes, through stimulation by parasites, and cytokines from damaged hepatocytes. These specimens were prepared from 40 cystic echinococcosis patients [40].

### 3.4. Other Hepatic Fibrosis Factors Induced by Echinococcus

Th1-type cytokines promote cell recruitment around larva and participate in chronic cell infiltration, leading to a dense granuloma that eventually progresses into fibrosis and necrosis. Th1 immune responses are the main cause of irreversible fibrosis in the host. The experiments were performed in a murine alveolar echinococcosis model, as well as in human alveolar echinococcosis patients [41].

Another study showed that TNF-α and/or LT-α genes play an essential role in immune protection mechanisms against *E. multilocularis* at the site of infection in mice model [42].

MMPs (matrix-metallo-proteinases) are zinc-dependent proteases secreted as prozymogen, and are activated to degrade the ECM. In the liver of mice infected with *E. granulosus*, the expression of MMP2 is high in early post infection and decreases in the middle and late post infection, suggesting that early inflammatory cells secrete proteases including MMP2, reducing the deposition of the ECM. This may be related to the innate immune response. In the late post infection, the expression of MMP2 decreases gradually, and the ECM deposits in large quantities and hepatic fibrosis further develops. The overexpression of MMP2 also leads to the over-activation of HSCs, initiating hepatic fibrosis [43].

Other investigators have found that mammalian FGF, present in the liver and upregulated during fibrosis, could induce the metacestode of *E. multilocularis* to grow during alveolar echinococcosis by acting on an evolution conserved parasite FGF signaling system (parasite FGF receptors). Growth of metacestodes could stimulate hepatic fibrosis as a vicious circle is created. They also reported that the parasite’s FGF signaling systems are promising targets for the development of novel drugs against alveolar echinococcosis. These experiments showed that tyrosine kinase (TK) inhibitor BIBF 1120, which is used to treat human cancer, could prevent activation of FGF signalling system. In vitro experiments were performed with the natural metacestodes of *E. multilocularis* (from rodents). Metacestodes were then continuously passaged in *mongolian jirds* (*Meriones unguiculatus*) [44].

## 4. Fibrosis in Other Organs Induced by *Echinococcus*

Cystic echinococcosis in bone is quite rare (accounting for 0.5–2% of all cases, half of which are in the spine) [45,46,47]. A case of a sacral hydatid cyst was reported as a more uncommon case. The patient was a 19 years old man presenting with cauda equine syndrome and with pelvic pain as a result of constriction from a hydatid cyst with fibrous tissue around it. After surgical resection, the evaluation was good, with recovery of perineal sensation and anal tone. Albendazole is the drug of choice against this disease. When suspected, presurgical use of Albendazole in cystic echinococcosis reduces risk of recurrence and/or facilitates surgery by reducing intracystic pressure. A misdiagnosis of hydatid cyst could be devastating. Hence, hydatid cyst should be kept as a differential diagnosis and when encountered with a cystic lesion of sacrum. In addition, long term follow-up is mandatory as risk of recurrence is high [48].

It has been found that hydatid cyst fluid of *E. granulosus* could stimulate epithelial to mesenchymal transition in human adenocarcinoma-derived alveolar basal epithelial cell line (A549) when cultured in vitro. It has also been reported that hydatid cyst fluid of *E. granulosus* could enhance expression of mesenchymal cytoskeletal markers (fibronectin and vimentin) accompanied by a down-regulation of an epithelial marker (E–cadherin gene) in human adenocarcinoma derived alveolar basal epithelial cell line. This indicates that epithelial to mesenchymal cell transition induced by *E. granulosus* could contribute to fibrotic reactions in lung tissue [49].

In humans, fibrosis in the pericardium induced by *Echinococcus* may cause consequent pericardial effusion, acute pericarditis, cardiac tamponade or constrictive pericarditis. Additionally, intracardiac rupture of an *Echinococcus* cyst could cause membrane or secondary cyst embolization in the lungs or organs supplied by the systemic circulation. Although unusual causes of cardiac disease, heart involvement by *Echinococcus* should be considered in differential diagnosis, especially of myocardial and/or pericardial diseases of unknown etiology, in both immunocompetent and immunocompromised individuals [50].

## 5. Therapeutics

The restoration of the liver structure has been reported in mice infected by *E. granulosus* after treatment of 125 pg/mLIL-17A without any visible toxic effects. Lower fibrosis and reduced iNOS, TNF-α, NF-κb and CD68 expression in the hepatic parenchyma of treated mice were also noted [51].

With knowledge of the collagen fiber surrounding the metacestodes of *E. granulosus*, others have discovered a separable space between the inner and outer layers of fibrocysts stratified round the metacestodes [45]. Xiangmo and colleagues surgically removed more than 100 cases of hydatid cysts (metacestodes) along potential separable spaces between the inner and outer layers of patients infected with *E. granulosus*. Successful surgical operation can treat patients and remove the metacestodes of *E. granulosus* completely without damaging the surrounding ductal tissues (e.g.*,* blood vessels and bile ducts) and metacestodes, avoiding recurrence and complications of the residual cavity post-operation [52,53].

Labsi and coworkers reported that the combination of pomegranate peel aqueous and albendazole showed more hydatid cyst growth inhibition; the histological structure of liver improvement and significant iNOS, TNF-α, NF-κβ, vimentin, CD68 decrease expression compared with albendazole-treated groups in mice infected with *E. granulosus* [54].

## 6. Conclusions

In recent years, miRNA and TGF-β have been hot spots with respect to investigating the mechanisms by which hepatic fibrosis is induced by *Echinococcus* spp. A thorough understanding of these mechanisms would be conducive to drug development. Morphological studies of the metacestode of *Echinococcus* spp. and the fibrocysts surrounding these tissues may lead to a deeper understanding of their generation, and may promote surgical methodologies and other possible interventions.

Still, the molecular mechanisms of parasitic organ tropism are little studied to date. Also, most studies have been focused on the signaling pathways in the host cells. The stimulation from *Echinococcus* spp. has usually been described as ‘excretion/secretion products’, but not as ligand-specific. Little research has been concerned with how host components interact with *Echinococcus* spp. and the signaling pathways in the parasite. To clarify and confirm the complete process of the *Echinococcus*–host interaction, additional studies are required to focus on the parasitic side as well as their involvement in this complex process.

## Figures and Tables

**Figure 1 diseases-07-00051-f001:**
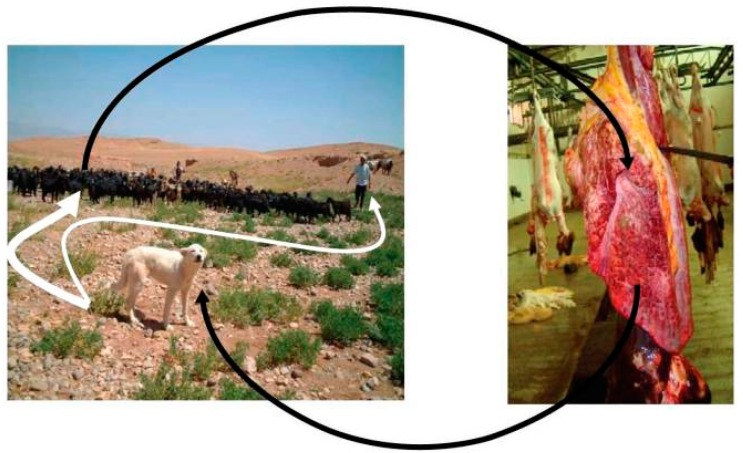
Life cycle of *E. granulosus* [6]. Copyright notice: The link of original article: (https://journals.plos.org/plosntds/article?id=10.1371/journal.pntd.0001146), the link of the Creative Commons Attribution License (http://creativecommons.org/licenses/by/2.0), if any changes were made in the original picture: □Yes ☑No.

**Figure 2 diseases-07-00051-f002:**
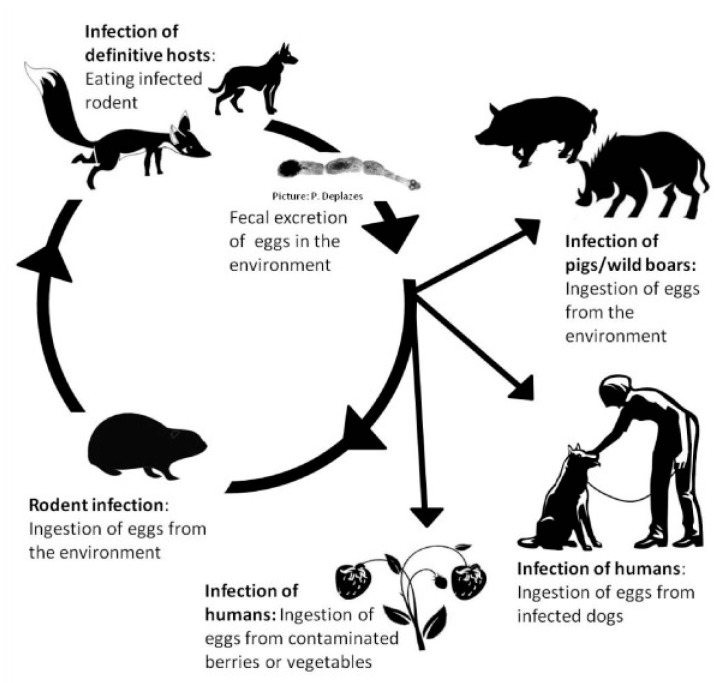
Life cycle of *E. multilocularis* [7]. Copyright notice: The link of original article: (https://actavetscand.biomedcentral.com/articles/10.1186/1751-0147-53-9), the link of the Creative Commons Attribution License (http://creativecommons.org/licenses/by/2.0), if any changes were made in the original picture: □Yes ☑No.

**Figure 3 diseases-07-00051-f003:**
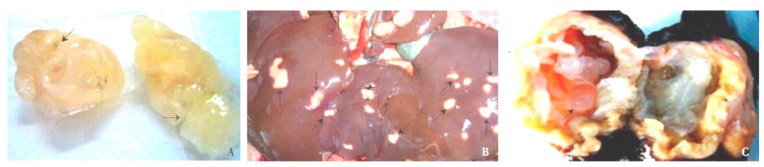
(**A**) Metacestodes of *E. multilocularis* in rodents with daughter cysts (→). (**B**) Uneven surface of the liver containing about 20 metacestodes of *E. granulosus* (→). (**C**) Typical metacestodes of *E. granulosus* with daughter cysts (→) in livers of sheep. Cyst walls showing extensive fibrosis.

**Figure 4 diseases-07-00051-f004:**
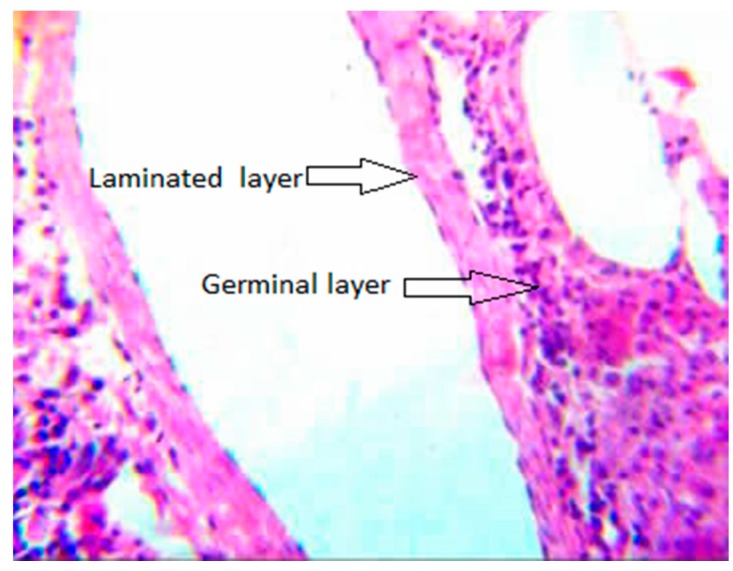
Liver pathology of sheep infected by *E. granulosus* (×400). Note the laminated layer (→) with a glue peptone like appearance in the germinal layer (→).

**Figure 5 diseases-07-00051-f005:**
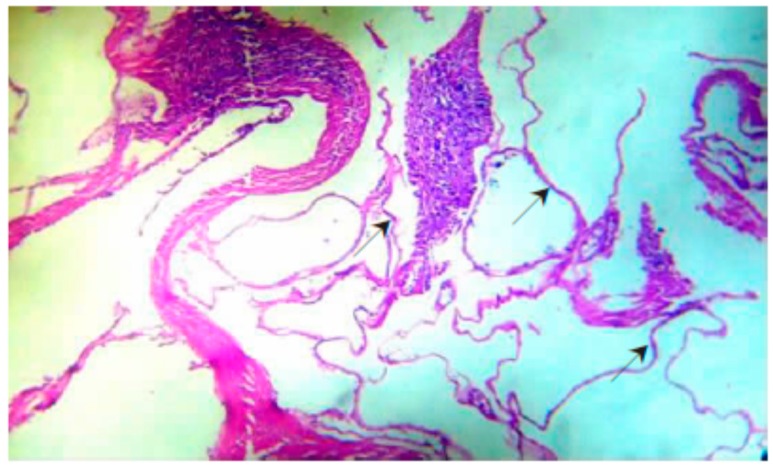
Liver pathology of the rodents infected with *E. multilocularis* (×400). Note: numerous metacestodes gather in groups, with germinal and laminated layers. laminated layers dropped from germinal layers. The laminated layers showing belt-shaped without cell structure curling in the cyst (→).

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
