# Peer review of "Mechanism of Fibrosis Induced by Echinococcus spp."

_diseases, 2019, doi:10.3390/diseases7030051_

Round 1
Reviewer 1 Report
Dear Authors
Concerning your manuscript, I believe it is an interesting issue at a specific and broad level of human and comparative pathology of a very important zoonotic group of agents at human level and this type of review should have more visibility.
The text must be improved in many details concerning grammar and the review of many typos.
Besides what will be pointed out in my detailed review, namely that your manuscript must be thoroughly reviewed to be published, the final decision on the publication of your manuscript depends on the Editor final statement.
Regarding my reviews and comments, they are as follows:
Title
When you mention Echinococcus, you really meant all and every species of this genus? Because the way they develop and expand the larval stages in the liver is quite different and so, it is different the inflammation regarding its presence. Please specify better what you meant.
Abstract
Page 1/5
Line 8 – Hepatic should not be in bold. Also in this line, you must separate these two words Echinococcusinfection. This kind of typo is very frequent all over the text (for instance, hypertensionetc in line 9), so I won’t name it again, but you must proceed to a thorough analysis of all the English and typos/mistakes in the manuscript.
Line 13 – Write Echinococcus in italic. Also, give space after a dot and correct “…treatment.Research…”.
1. Introduction
Page 1/5
Line 19 – Instead of “EchinococcusSpp.Include E. granulosus, E. multilocularis, E. Vogeli, E.oligarthrus and E.shiquicus.” write “Echinococcus spp. include E. granulosus, E. multilocularis, E. vogeli, E. oligarthrus and E. shiquicus. And the expressions sp. and spp. are never written with italic, but with normal font. And always give one space between the genus and species names.
Lines 23-25 – You must not generalize, because this statement refers only to E. multilocularis, so you must mention it in the text.
Line 34 – “hepatic stellate cells” should be written with the same letter as the rest of the text, e.g. in Times New Roman.
2. The Mechanismof Hepatic Fibrosis induced by Echinococcus.
Separate “Mechanismof” and remove the dot at the end of the chapter.
Line 52 – Instead of “…, and was…” write “…, it was…”.
Line 80 – What is the number for the reference Xiangmo Wu et al.?
Line 96 – Instead of “Inner” write “inner”.
3. Methodology
Page 3/5
Write the title with a first capital letter
You write “I collected informations from pubmed and CNKI.” But, you should have written “We collected information from PubMed and CNKI”.
Besides, I think you should swap this chapter with the former, so that we understand better the way you performed your study before you develop it.
Besides, and being a search performed in two databases, you should inform the readers on the key words used, the range of the search (for example, 2000-2019?) and any other information that you find suitable, namely how did you filter the papers you used for the review and how did you reject the remaining ones. This also makes part of the methodology and you should explain it thoroughly.
4. Conclusion
You talk about development of new drugs and improvement in surgical methodology as two main consequences regarding your conclusions. Nevertheless, you should also mention something that has been done in the last 10-15 years regarding these issues, since it will explain better why you decided to perform this review and what are the main gains on having a better understanding on the genesis and structure of the liver fibrosis induced by Echinococcus spp.
As final comments, I have the following ideas:
I missed a scheme or figure, as well as macro and microscopic photos regarding the reviewed larval stages.
Finally, I recommend that in each statement regarding any mechanism or genesis of fibrosis, you remark which host are you talking about, namely if it was studied after experimental or natural infections of rodents or sheep (or other domestic animal) and in natural infections regarding human cases.
5. References
Pages 3-5/5
Insert a number before the chapter, for the sake of coherency and write it in plural like I did.
Please check if each, and every reference, is cited in the text and vice-versa.
All scientific names mentioned in papers’ titles must be in italic.
Best regards and good luck with your revision.
Reviewer 1
Author Response
Response to Reviewer 1 Comments
Point 1: Concerning your manuscript, I believe it is an interesting issue at a specific and broad level of human and comparative pathology of a very important zoonotic group of agents at human level and this type of review should have more visibility.
Response 1: Please notice additional figures which have been added.
Point 2: When you mention Echinococcus, you really meant all and every species of this genus? Because the way they develop and expand the larval stages in the liver is quite different and so, it is different the inflammation regarding its presence. Please specify better what you meant.
Response 2: As recommended the title has been changed.
Point 3: Abstract, Page 1/5 ,Line 8 – Hepatic should not be in bold. Also in this line, you must separate these two words Echinococcusinfection. This kind of typo is very frequent all over the text (for instance, hypertensionetc in line 9), so I won’t name it again, but you must proceed to a thorough analysis of all the English and typos/mistakes in the manuscript.
Response 3: It’s been revised.
Point 4: Line 19 – Instead of “EchinococcusSpp.Include E. granulosus, E. multilocularis, E. Vogeli, E.oligarthrus and E.shiquicus.” write “Echinococcus spp. include E. granulosus, E. multilocularis, E. vogeli, E. oligarthrus and E. shiquicus. And the expressions sp. and spp. are never written with italic, but with normal font. And always give one space between the genus and species names.
Response 4: It’s been revised.
Point 5: Lines 23-25 – You must not generalize, because this statement refers only to E. multilocularis, so you must mention it in the text.
Response 5: It’s been revised.
Point 6:Line 34 – “hepatic stellate cells” should be written with the same letter as the rest of the text, e.g. in Times New Roman.
Response 6: It’s been revised.
Point 7: The Mechanismof Hepatic Fibrosis induced by Echinococcus. Separate “Mechanismof” and remove the dot at the end of the chapter.
Response 7: It’s been revised.
Point 8: Line 52 – Instead of “…, and was…” write “…, it was…”.
Response 8: It’s been revised.
Point 9: Line 80 – What is the number for the reference Xiangmo Wu et al.?
Response 9: It’s been added
Point 10: Line 96 – Instead of “Inner” write “inner”.
Response 10: It’s been revised.
Point 11: Write the title with a first capital letter
Response 11: It’s been revised.
Point 12: You write “I collected informations from pubmed and CNKI.” But, you should have written “We collected information from PubMed and CNKI”. Besides, I think you should swap this chapter with the former, so that we understand better the way you performed your study before you develop it. Besides, and being a search performed in two databases, you should inform the readers on the key words used, the range of the search (for example, 2000-2019?) and any other information that you find suitable, namely how did you filter the papers you used for the review and how did you reject the remaining ones. This also makes part of the methodology and you should explain it thoroughly.
Response 12: We have revised the paper according and extensively extended the methodology as reviewer has suggested.
Point 13: You talk about development of new drugs and improvement in surgical methodology as two main consequences regarding your conclusions. Nevertheless, you should also mention something that has been done in the last 10-15 years regarding these issues, since it will explain better why you decided to perform this review and what are the main gains on having a better understanding on the genesis and structure of the liver fibrosis induced by Echinococcus spp.
Response 13: As reviewer suggested we have added addition material.
Point 14: I missed a scheme or figure, as well as macro and microscopic photos regarding the reviewed larval stages.
Response 14: It’s been added.
Point 15: Finally, I recommend that in each statement regarding any mechanism or genesis of fibrosis, you remark which host are you talking about, namely if it was studied after experimental or natural infections of rodents or sheep (or other domestic animal) and in natural infections regarding human cases.
Response 15: It’s been added.
Point 16: Insert a number before the chapter, for the sake of coherency and write it in plural like I did. Please check if each, and every reference, is cited in the text and vice-versa. All scientific names mentioned in papers’titles must be in italic.
Response 16: It’s been added.
Reviewer 2 Report
The topic of the manuscript could be interesting, however, in my opinion this shouldn’t be classify as Review Article, it is at most a mini review. Authors’ revised only 32 articles among these only selected are directly related to the topic of hepatic fibrosis induced by Echinococcus. This topic is to shallow to be recognized as necessary to be revised, moreover, a review paper should include additional information or conclusions providing new points of view, clarifying and/or combining facts or showing new paths. These are missing in the present paper.
Because the article is related to a subject to which the available literature is easily accessible and easy to find, I do not see the need of a review article about this particular and limited topic. To help the Authors with future submissions I suggest to extend the paper, and discuss all of the pathological changes developed in the liver and the other tissues during echinococcal diseases and clarify their impact on the organ as well as clinical consequences for the host.
I do not recommend publication of this mini review in its current stage.
Minor comments:
Some sentences have been inserted into the content in isolation from the context and without any conclusions (see lines 47-59; 95-96; 107-108 etc.)
References format should be unified
Author Response
Response to Reviewer 2 Comments
Point 1: The topic of the manuscript could be interesting, however, in my opinion this shouldn’t be classify as Review Article, it is at most a mini review. Authors’ revised only 32 articles among these only selected are directly related to the topic of hepatic fibrosis induced by Echinococcus.
Response 1: We appreciate the interest and constructive comments to improve this paper. As suggested we have lengthened the paper and added additional related references now total 52.
Point 2: Reviewer suggests: revised, moreover, a review paper should include additional information or conclusions providing new points of view, clarifying and/or combining facts or showing new paths. These are missing in the present paper.
Response 2: It’s been added.
Point 3: Reviewer questions the need for this review paper due to the limitation of the topic.
Response 3: Please refer to the revised manuscript with additional material to improve the paper and prove the need for this review paper, as well as references 1,4,6,10,25,36,37,38,41,50,51.
Point 4: Reviewer suggests to extend the paper, and discuss all of the pathological changes developed in the liver and the other tissues during echinococcal diseases and clarify their impact on the organ as well as clinical consequences for the host.
Response 4: As suggested we have added Fibrosis in other organs induced by Echinococcus, and present data on fibrosis as the focus of this paper and extended pathological changes in other organs.
Point 5: Some sentences have been inserted into the content in isolation from the context and without any conclusions (see lines 47-59; 95-96; 107-108 etc.)
Response 5: Tx for the comments we have extensively revised the manuscript.
Point 6: References format should be unified
Response 6: It’s been done
Thanks for all your constructive comments and attention to our revised manuscript. Please don’t hesitate to suggest more comments to improve our manuscript. We are looking forward for your decision.
Sincerely Yours
Authors

Reviewer 3 Report
The paper is faced with many errors starting in the first line of the abstract.
This paper doesn't full fill the criteria to be published.
Author Response
Response to Reviewer 3 Comments
Point 1: The paper is faced with many errors starting in the first line of the abstract.
This paper doesn't full fill the criteria to be published.
Response 1: We agree and the manuscript has been extensively revised.
Thanks for all your constructive comments and attention to our revised manuscript. Please don’t hesitate to suggest more comments to improve our manuscript. We are looking forward for your decision.
Sincerely Yours
Authors
Round 2
Reviewer 1 Report
Dear Authors
Concerning your manuscript, I continue to believe it is an interesting issue at a specific and broad level of human and comparative pathology of a very important zoonotic group of agents at human level and this type of review should have more visibility. You did a good revision in your manuscript, which was improved in many details, namely grammar and typos reviews with yellow highlights and even with original figures to better illustrate your work. Moreover, you addressed all my comments and followed my advices.
Besides what will be pointed out in my second review, namely that your manuscript now has more potential to be published, the final decision on the publication of your manuscript depends on the Editor final statement.
Regarding my reviews and comments, they are as follows:
Title
When you use the expression Echinococcus Spp., you should write it Echinococcus spp.
Authors
Check if every author’s name is correctly written.
Introduction
Page 1/10
Line 25 - Write Echinococcus spp. and E. vogeli.
Lines 28 – 29 – Write ”… cattle, sheep and other herbivores or humans…” without italic.
Line 31 – Write within
Line 33 – Write dogs without italic. From now on, this rule applies to every common name written in italic. You must use italic only for scientific names.
Line 36 – Write Felis.
Line 39 – Write “…shown in Fig. 1 and Fig. 2, respectively…”
Line 42 – Instead of migrant write migrate.
Page 2/10
Line 46 – Write differentiation
2. Mechanisms of Hepatic Fibrosis induced by Echinococcus
Page 4/10
Figure 4 – Differentiate better the place pointed by each arrow regarding the laminated layer and germinal layer.
Figure 5 title – What do you mean by “…with germinal and laminated layers (→) that are separate from germinal layers.”?
Page 5/10
Line 126 - Write lesions [26].
Page 6/10
Line 167 – Write model [40].
Line 183 – Write E. multilocularis.
Line 202 – Write fibrotic.
Page 7/10
Line 224 – Instead of “…expression compeared…”, write “…expression compared…”
5. Methodology
First of all, I would place this chapter as the number 2, before the “Mechanisms of Hepatic Fibrosis induced by Echinococcus”, since your methodology drove the rest of your manuscript.
Line 229 – Instead of “…both tow databases…”, write “…both two databases…”.
Line 240 – Write “…according to routine procedures.”
Finally, and for the whole paper and bibliography, all scientific names mentioned in references papers’ titles must be in italic.
Best regards and good luck with your revision.
Reviewer 1
Author Response
Dear Reviewer 1,
We have revised our manuscript according the constructive comments of the reviewers as follow:
Response to Reviewer 1 Comments
Point 1: When you use the expression Echinococcus Spp., you should write it Echinococcus spp.
Response 1: It’s been revised
Point 2:Check if every author’s name is correctly written.
Response 2:It’s been done
Point 3: Line 25 - Write Echinococcus spp. and E. vogeli.
Response 3:It’s been revised
Point 4: Lines 28 – 29 – Write ”… cattle, sheep and other herbivores or humans…” without italic.
Response 4:It’s been revised
Point 5: Line 31 – Write within
Response 5:It’s been revised.
Point 6: Line 33 – Write dogs without italic. From now on, this rule applies to every common name written in italic. You must use italic only for scientific names.
Response 6:It’s been revised.
Point 7:Line 36 – Write Felis.
Response 7:It’s been revised.
Point 8: Line 39 – Write “…shown in Fig. 1 and Fig. 2, respectively…”
Response 8:It’s been revised.
Point 9: Line 42 – Instead of migrant write migrate.
Response 9:It’s been revised.
Point 10: Line 46 – Write differentiation
Response 10:It’s been revised.
Point 11: Figure 4 – Differentiate better the place pointed by each arrow regarding the laminated layer and germinal layer.
Response 11:It’s been revised.
Point 12: Figure 5 title – What do you mean by “…with germinal and laminated layers (→) that are separate from germinal layers.”?
Response 12:It’s been revised.
Point 13:ine 126 - Write lesions [26].
Response 13:It’s been revised.
Point 14: Line 167 – Write model [40].
Response 14:It’s been revised.
Point 15:Line 183 – Write E. multilocularis.
Response 15:It’s been revised.
Point 16: Line 202 – Write fibrotic.
Response 16:It’s been revised.
Point 17:Line 224 – Instead of “…expression compeared…”, write “…expression compared…”
Response 17:It’s been revised.
Point 18:First of all, I would place this chapter as the number 2, before the “Mechanisms of Hepatic Fibrosis induced by Echinococcus”, since your methodology drove the rest of your manuscript.
Response 18:It’s been done.
Point 19:Line 229 – Instead of “…both tow databases…”, write “…both two databases…”.
Response 19:It’s been revised.
Point 20:Line 240 – Write “…according to routine procedures.”
Response 20:It’s been revised.
Point 21:Finally, and for the whole paper and bibliography, all scientific names mentioned in references papers’ titles must be in italic.
Response 21:It’s been revised.
Reviewer 2 Report
I have no further comments on the manuscript.
Author Response
Response to Reviewer 2 Comments
Point 1:I have no further comments on the manuscript.
Response 1:Thank you so much.
Reviewer 3 Report
the MS has been improved but still not ready to be forwarded and accepted.
Many mistakes in the text. Only in the introduction, i noticed following orthographic mistakes, f.e.:
l.25 Spp/please, use small a ..spp.
l.26 E. Vogeli / small v
l.28 name of the animals: not cursive / and subsequently in the following text
l.47 diffferentiaion - differentiation
l.69, , Alveolar e....alveolar e.
figs. keep space between text / species nams and lit no reference.
what is the origin of the images? sources?
some images are original fotos some other from the internet. please, provide exactly the source.
2.
2.1. Headlines/ ....Hepatic Fibrosis/ please, use h and f
References
most of the references are not correctly presented/style of citation
f.e. please, have a look in nr 3 + 4
Author Response
Response to Reviewer 3 Comments
Point 1:l.25 Spp/please, use small a ..spp.
Response 1:It’s been revised.
Point 2:l.26 E. Vogeli / small v
Response 2:It’s been revised.
Point 3:l.28 name of the animals: not cursive / and subsequently in the following text
Response 3:It’s been revised.
Point 4:l.47 diffferentiaion - differentiation
Response 4:It’s been revised.
Point 5:l.69, , Alveolar e....alveolar e.
Response 5:It’s been revised.
Point 6:figs. keep space between text / species names and lit no reference.
Response 6:It’s been revised.
Point 7:what is the origin of the images? sources?
Response 7: Tx we have explained in the Methodology part. Fig 1 and Fig 2 have been changed, Fig 1-2 were taken from PubMed database by searching ‘lifecycle of E. granulosus’ and ‘lifecycle of E. multilocularis.’ Fig 3-5 are from our group.
Point 8:some images are original fotos some other from the internet. please, provide exactly the source.
Response 8: Tx we have explained in the Methodology part. Fig 1 and Fig 2 have been changed, Fig 1-2 were taken from PubMed database by searching ‘lifecycle of E. granulosus’ and ‘lifecycle of E. multilocularis.’ Fig 3-5 are from our group.
Point 9:2.1. Headlines/ ....Hepatic Fibrosis/ please, use h and f
Response 9:It’s been revised.
Point 10:most of the references are not correctly presented/style of citation
f.e. please, have a look in nr 3 + 4
Response 10: Tx for the comment, we have revised and corrected the references.